# AI-Enabled Intelligent Visible Light Communications: Challenges, Progress, and Future

**Jianyang Shi** [1,2,3] , **Wenqing Niu** [1] , **Yinaer Ha** [1] , **Zengyi Xu** [1] , **Ziwei Li** [1,4] , **Shaohua Yu** [4] **and Nan Chi** [1,2,3,*]

1 Key Laboratory for Information Science of Electromagnetic Waves (MoE), Fudan University, Shanghai 200433, China; jy_shi@fudan.edu.cn (J.S.); 21110720066@m.fudan.edu.cn (W.N.); 19210720063@fudan.edu.cn (Y.H.); 21110720075@m.fudan.edu.cn (Z.X.); lizw@fudan.edu.cn (Z.L.)
2 Shanghai Engineering Research Center of Low-Earth-Orbit Satellite Communication and Applications, Shanghai 200433, China
3 Shanghai Collaborative Innovation Center of Low-Earth-Orbit Satellite Communication Technology, Shanghai 200433, China
4 Peng Cheng Laboratory, Shenzhen 518055, China; shaohuayu@fudan.edu.cn
* Correspondence: nanchi@fudan.edu.cn

**Abstract:** Visible light communication (VLC) is a highly promising complement to conventional wireless communication for local-area networking in future 6G. However, the extra electro-optical and photoelectric conversions in VLC systems usually introduce exceeding complexity to communication channels, in particular severe nonlinearities. Artificial intelligence (AI) techniques are investigated to overcome the unique challenges in VLC, whereas considerable obstacles are found in practical VLC systems applied with intelligent learning approaches. In this paper, we present a comprehensive study of the intelligent physical and network layer technologies for AI-empowered intelligent VLC (IVLC). We first depict a full model of the visible light channel and discuss its main challenges. The advantages and disadvantages of machine learning in VLC are discussed and analyzed by simulation. We then present a detailed overview of advances in intelligent physical layers, including optimal coding, channel emulator, MIMO, channel equalization, and optimal decision. Finally, we envision the prospects of IVLC in both the intelligent physical and network layers. This article lays out a roadmap for developing machine learning-based intelligent visible light communication in 6G.

**Keywords:** visible light communication; artificial intelligence; machine learning; physical layer; network layer

## 1. Introduction

As 5G's commercialization progresses, the number of 5G base stations worldwide has surpassed one million. This marks the beginning of globally competitive future-oriented research on 6G networks. According to several research reports [1–3], it is widely assumed that 6G communication will go beyond the current wireless spectrum and shift towards higher frequencies. The millimeter-wave and terahertz spectrum have long been the research focus academically and industrially, except that the equipment is of extremely high cost. Recently, the spectrum of light, i.e., visible and infrared light, provides a potential supplement for 6G. During the last decade, visible light communication is being cast in the spotlight by 6G researchers as a green, energy-efficient, high-speed communication method [4].

Visible light communication transmits (VLC) signals in a spectrum range of 400–800 Thz, which owns a very different physical property compared with both conventional wireless transmission and optical communication. Communication with visible light provides benefits of electromagnetic interference resistance, vast spectrum resources, and high-speed transmission capabilities. Moreover, it can be equipped with common lighting systems

to allow simultaneous illumination and communication. Furthermore, the short wavelength of light source allows for the creation of super-compact cells, which are ideal for 6G communication. Nevertheless, signal communication at such a small wavelength poses critical challenges to transmitting and receiving devices. Semiconductor materials with wide bandgaps must be employed to achieve such high-frequency photons [5]. The extra electro-optical and photoelectric conversions compared to wireless communications introduce undesirable nonlinear distortions and hinder the high-speed transmission in visible light communications [6,7]. Traditional algorithms and strategies can help to mitigate the specific negative influence from visible light to its communication performance [8,9]. However, these algorithms cannot offset the performance difference between VLC applications and their existing counterparts. Thankfully, artificial intelligence (AI) has become a critical component of the 6G network [10]. It is expected to be the optimal solution for enabling visible light communication.

Machine learning (ML) has emerged to be the most popular technique for prediction, classification, and pattern identification, and has shown great success in data mining, image recognition, and other areas in the last decade. The recent development of AI processing units further accelerates the advancement of the more powerful deep neural networks (DNN). Many machine learning techniques have been successfully implemented in the fields of optical communication [11] and wireless communication [12]. However, the machine learning algorithm also has their own set of drawbacks, such as high computational complexity, long training times, and poor generalization. In the more complicated visible light communications, these issues will be amplified. In the more complicated visible light communications, these issues will be amplified. Therefore, machine learning should be wisely adopted to the visible light communication scenario, in the case that it may not be a viable solution.

Nowadays, wireless networks have progressed from software-defined radio (SDR) and cognitive radio (CR) [13] to AI-powered intelligent radio (IR) [10]. Visible light communication, as a communication method sprouting from 6G, aims to skip the first two stages and go directly to the IR stage. To accomplish this leap, we need to build the framework of intelligent visible light communication (IVLC). IVLC will be a broad concept covering both the intelligent physical layer and the intelligent network layer (including the traditional data link layer and network layer). As we have seen, 6G is still in its early stages of development, and 6G-based IVLC is in an even more preliminary stage. Therefore, the intelligent physical layer, which is more different from traditional wireless, could be the core breakthrough point in forthcoming years.

In this paper, we will introduce the concepts of the intelligent physical layer and the intelligent network layer of IVLC. The underlying physical layer will be given great consideration. Among the existing machine learning algorithms, there are four main categories according to the purpose of implementation: regression [14], classification [15], clustering [16], and dimensionality reduction [17]. However, in IVLC, especially in the physical layer, the existing machine learning algorithm is not designed to achieve the above functions. For this reason, we redefine the categories of machine learning techniques in the physical layer of the IVLC based on the communication system framework, including optimal coding, channel emulator, MIMO, channel equalization, and optimal decision. As seen in Section 3, such categorization intersects with the traditional ML applications, which facilitates readers who are interested in investigating intelligent visible light communication. Each module of the communication framework is featured with unique issues and thus requires specially-designed machine learning algorithms. We will go through the major obstacles of visible light communication and discuss how AI-empowered IVLC could overcome them. It is possible that the newly emerging intelligent visible light communication may play a key role in the 6G communication network, enabling worldwide smart connectivity and the construction of air–space–ground–sea integrated networks.

## 2. Statues and Challenges of VLC

Visible light communication introduces special nonlinearities due to the additional electro-optical conversion, which can significantly impair communication performance. Due to the spontaneous radiation properties of LEDs, visible light signals can only be directly modulated in communication. This means that changes in signal amplitude will directly affect the carrier concentration, and thus the recombination of electrons and holes [18]. In this section, we present the complexity of visible channels in terms of physical channels and modulation formats, and afterward, show the superiority of machine learning and the attendant costs.

### 2.1. Visible Light Communication E2E Channel

The VLC end-to-end channel $\mathcal{H}(x)$ includes a digital-to-analog converter (DAC, in arbitrary waveform generator), electronic amplifier (EA), bias tee, LED, transmission channel, receiver, and analog-to-digital converter (ADC, in oscilloscope), as shown in Figure 1. The entire transmission link contains electrical voltage signals, current signals, and optical signals, as well as the conversion between them. However, due to the complexity of the visible light channel, research now focuses on the LED emitter and the transmission channel, which are (d), (e), and (f) of Figure 1.

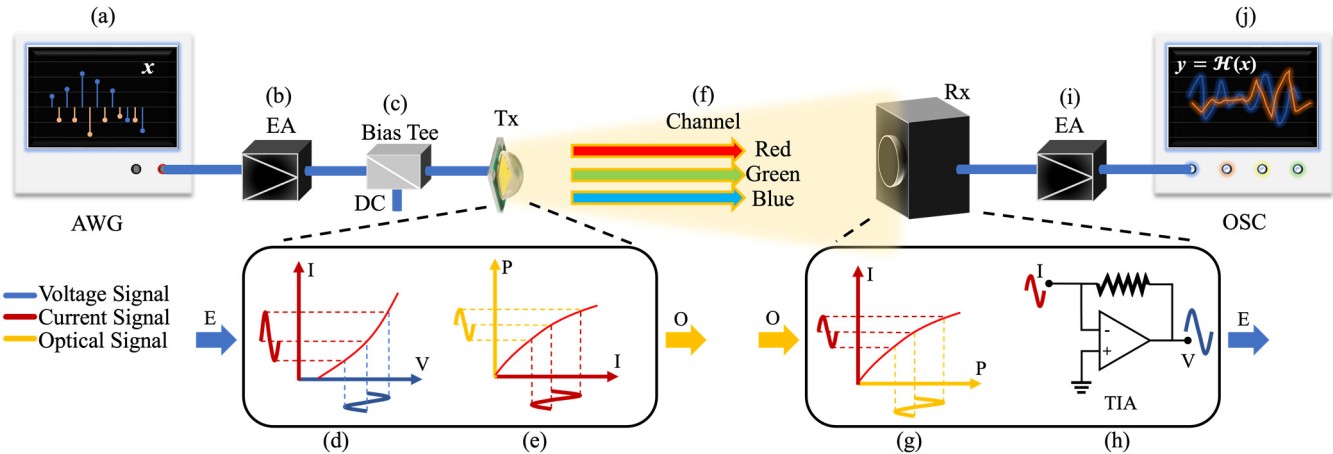

**Figure 1.** Overview of VLC end-to-end channel; (**a**) DAC, (**b**) EA, (**c**) bias tee, (**d**) V-I, (**e**) I-P, (**f**) transmission channel, (**g**) P-I, (**h**) transimpedance amplifier (TIA), (**i**) EA and (**j**) ADC.

From the communication point of view, the most primitive LED transmission model is the frequency domain model, which is given as [19]:

$$H(\omega) = e^{-\frac{\omega}{\omega_c}} \tag{1}$$

where $\omega_c$ is a fitted coefficient. This model mainly represents not only the frequency-selective fading phenomenon of the visible light channel, but also the inter-symbol interference (ISI) and linear memory effects. The high-frequency fading of the exp (exponential) fits well with the limited bandwidth of visible light communication. Therefore, when only linear channels are considered, the expression form of exp is also applicable to the underwater visible light channel [20].

However, a linear model alone cannot describe a channel that is as complex as visible light communication. The first consideration is the V-I transfer model of VLC [21], which extends from the solid state power amplifier (SSPA) model [22]. Similarly, there are equivalent circuit models [23] to equate the V-I transfer curve and frequency response of VLC. Such a model implies that the nonlinear term is only amplitude-dependent, independent of frequency and time, and independent of adjacent symbols.

If we want to take all the above factors into account, a simple way is to equate the overall linearity and nonlinearity to the Volterra series [24,25], as a black-box model. A second-order Volterra expansion can be expressed as [25]:

$$y(m) = \sum_{i=0}^{N-1} h(i)x(m-i) + \sum_{i=0}^{L-1}\sum_{j=k}^{L-1} w(ij)x(m-i)(m-j) \tag{2}$$

where $h$ and $w$ are the linear and nonlinear weights, and $N$ and $L$ are the tap numbers of linearity and nonlinearity. When the signal bandwidth is small (14 MHz bandwidth) and the signal amplitude is small (40 mA DC bias), the second-order Volterra is proven to have better similarity [24]. High-speed visible light communication, however, often requires a higher signal-to-noise ratio (SNR) and modulation bandwidth to meet the 6G transmission rate requirements. Higher-order Volterra series may be expected to fit well, but would introduce an exponential increase in computational complexity.

None of the above-mentioned channel models from the communication dimension actually take the substance of photoelectric and electro-optical conversion into account in computation. They simply treat it as a black box. Since LED is a GaN wide-bandgap semiconductor material based on a multi-quantum wells (QWs) structure, the carrier rate equation can be modeled for LED from semiconductor material considerations [26–29]. The model is named as the ABC model, where A, B, and C represent Shockley–Read–Hall (SRH, nonradiative) recombination, radiative recombination, and Auger recombination (nonradiative), respectively. Considering both recombination and leakage of carriers, the recombination rate $R$ can be expressed as follows [28]:

$$R = An + Bn^2 + Cn^3 + f(n) \tag{3}$$

$$f(n) \approx an + bn^2 + cn^3 + dn^4 \ldots \tag{4}$$

where $A$, $B$, and $C$ represent the SRH, radiative, and Auger recombination coefficient. $n$ is the minority carrier concentration. $f(n)$ is the carrier leakage term, which has been expanded into the Taylor series. The carrier lifetime $\tau$, which determines the modulation bandwidth, is given as [18]:

$$\tau = \frac{n}{R} \approx \frac{n}{An + Bn^2 + Cn^3 + an + bn^2 + cn^3 + dn^4 \ldots} \tag{5}$$

The carrier density is determined by the effective injected current density, which is expressed as [27]:

$$\frac{\Delta n}{\Delta R} = -R + \frac{\eta_{inj}J}{q_e} \tag{6}$$

$$J = \frac{I}{w_{active}} \tag{7}$$

where $\eta_{inj}$ is the injection efficiency, $J$ is the current density, $q_e$ is the elementary charge, $I$ is the current intensity, and $w_{active}$ is the thickness of the effective active region. Then, the optical output power is given by [18]:

$$P = V_{active}E_{phot}\eta_{EQE}Bn^2 \tag{8}$$

here, $V_{active}$ is the volume of the active layer, $E_{phot}$ is the photon energy and $\eta_{EQE}$ is the external quantum efficiency (EQE).

As can be seen here, since the initial transmission signal is in the form of a voltage; it first goes through a nonlinear V-I conversion [21]. Then, the relationship between current and carriers is a dynamic nonlinear relationship. Moreover, at higher currents, there will be an efficiency droop [28]. It is also easy to understand that the effective radiative carriers have only second-order terms and the total number of carriers has higher-order terms.

When the current increases, the number of carriers increases, and the effective light-emitting carrier ratio $P_{ratio}$ will first increase and then decrease.

$$P_{ratio} = \frac{Bn^2}{An + Bn^2 + Cn^3 + an + bn^2 + cn^3 + dn^4 \ldots} \tag{9}$$

Because of the aforesaid dynamic nonlinear equations, all of the carrier-related publications mentioned above use a tiny signal and low bandwidth assumption in their derivation. The carrier density is minimal enough to ignore the higher-order terms due to the low current density caused by the tiny signal. If the signal has a limited bandwidth, the pulse duration is sufficient to make the left side of Equation (6) equal to zero. Abandoning the higher-order and differential terms would greatly simplify the modeling of visible light channels, but it also poses the problem of not being able to satisfy the modeling of high-speed VLC. In later work based on both the carrier rate equation and an equivalent discrete-time circuit modeling [7,30], the same assumptions were required, and the experimental verification of the channel modeling was performed with only 2 MHz [30].

In addition to the large signals and high bandwidth that make visible light channel modeling difficult [19,21], another challenge is transmission channel modeling [31–33]. Much work has focused on indoor visible light transmission modeling, such as multipath impulse-response analysis [31], ray-tracing methods based on channel impulse response [32], and photon-based statistical modeling [33]. However, this is only one aspect of visible light communication applications. When transmitting in an outdoor environment, the effects of atmospheric turbulence in visible wavelengths must also be modeled. Water environment modeling is also essential while broadcasting underwater.

Furthermore, as illustrated in Figure 1, numerous modules introduce nonlinearities. For example, there are high-power electrical amplifiers at the transmitter side, which can have large nonlinearities at high currents. The linear dynamic range of the receiver PIN is usually smaller than the LED, and too much optical power can cause saturation of the PIN, which is serious especially when using APD detectors. Therefore, the nonlinear modeling of the driver circuit and the receiver based on different detector implementations are also very important.

Because of the additional optoelectronic and electro-optical conversion, as well as the rest of the nonlinear modules, the VLC end-to-end channel is extremely complex, as summarized in Table 1. This also presents a significant problem for VLC's high-speed connectivity.

**Table 1.** Challenges of visible light communication E2E channel.

| Challenges | Reasons | References |
|---|---|---|
| **Optoelectronic and electro-optical conversion** | Introduces additional nonlinearity | [26–29] |
| **Large signals** | Brings the device into the nonlinear region | [21] |
| **Wide bandwidth** | Introduces severe ISI | [19] |
| **Different transmission channel modeling** | Diverse application scenarios, such as indoor, underwater | [31–33] |

## 2.2. Modulation Format in VLC

Because of the explained limitations in commercially available LED light sources, LED-based VLC system typically presents extremely limited bandwidth (several MHz). Apart from developing LEDs with novel structure and the optimization of the driving circuits, using advanced modulation formats is also an alternative for high-speed VLC systems. In this section, we will introduce several common modulation technologies in the VLC system.

On-off keying (OOK) as the most basic modulation format in a communication system, uses the "on" and "off" state of the carrier to transmit the binary information "1" and "0". The advantages of OOK modulation are simple implementation and low cost. In

2001, early research on LED-based VLC system applies an OOK non-return to zero (NRZ) modulation [34]. With the development of equalization technology, a 662-Mbit/s VLC link based on a single blue LED using OOK-NRZ modulation has been demonstrated [35].

Pulse amplitude modulation (PAM) is a one-dimensional (1-D) multilevel modulation. Compared with OOK, the spectral efficiency of PAM is less restricted. In [36], based on Volterra decision feedback equalization (DFE), a 1.1-Gbit/s white LED-based VLC system is experimentally demonstrated. Investigation on the comparison of the performance of PAM with different orders has also been reported in [37]. Experimental results indicate that through a three-tap pre-equalizer, a data rate of 2 Gbit/s is achieved.

For the VLC system, there is strong noise at the low-frequency components. Although this noisy spectrum can be avoided through up-conversion, the in-phase (I) and quadrature (Q) channels are not fully utilized if using 1-D modulation such as OOK or PAM. Carrierless amplitude-phase modulation (CAP) as a variant of QAM can not only avoid the low-frequency noise but also demonstrate a fuller utilization of the I and Q channels. It uses a pair of orthogonal Hilbert filters for up-conversion instead of subcarrier. CAP has been widely used in VLC systems due to its merits of low complexity and high spectral efficiency. The early demonstration of CAP modulation in VLC systems has been reported in 2012, in which a 1.1-Gbit/s 23-cm free space transmission is realized [38]. Multiband CAP has also been proposed for multi-user application; through flexible bit allocation, a VLC system with the spectral efficiency of 4.85 bit/s/Hz is demonstrated [39].

When using the above modulation technologies, equalizers with several taps are required to mitigate the ISI because of the bandwidth limitation effect in VLC system. If there is strong ISI, the taps of the equalizer will increase rapidly. Alternatively, multicarrier modulation technologies such as orthogonal frequency division multiplexing (OFDM), discrete multitone (DMT), and discrete Fourier transform spread (DFTS)-OFDM are possible to avoid ISI.

The OFDM signal is generated as follows: First, the transmitting sequence in the frequency domain is divided into parallel subchannels. Then, the time-domain symbols in a slot are the inverse fast Fourier transform (IFFT) of the frequency-domain symbols from each subcarrier. After adding CP and parallel to serial conversion, a complex-valued OFDM signal is generated. However, the transmitting signal is restricted to real value in VLC system. Therefore, an extra up-conversion is required for complex-to-real conversion. In [40], a 3-Gbit/s OFDM VLC system based on bit loading and power loading is demonstrated, indicating that OFDM has great potential of combining with adaptive bit- and power-allocation algorithms.

DMT is similar to OFDM, except that it uses Hermitian symmetry before IFFT, so that the signal after IFFT is real-valued. The step of up-conversion is not needed. In [41], using the maximum ratio combination a 2.3-Gbit/s underwater DMT VLC system is realized. Additionally, adaptive bit- and power-allocation algorithms can also be applied for DMT. It is reported that using a bit-loading and power-loading scheme, the data rate of the underwater VLC system based on Si-substrate LED has achieved 3.37 Gbit/s.

Although OFDM and DMT offer desirable resistance to ISI and flexible bit and power allocation, they are faced with a high peak-to-average power ratio (PAPR). DFTS-OFDM is proposed to mitigate the problem. The difference is that DFTS-OFDM employs an extra FFT operation between the serial-to-parallel conversion and IFFT. In [42], the authors have proven that by employing DFTS-OFDM the PAPR can be significantly reduced.

The complementary cumulative distribution function (CCDF) of the transmitted signal with different modulation formats is illustrated in Figure 2. The results indicate that the PAPR of OOK-NRZ is the lowest, followed by PAM 4. while the PAPR of CAP and PAPR of DFTS-OFDM are higher but exhibit similar performance. Obviously, OFDM has the highest PAPR. As a result, signals using different modulation may experience different nonlinear channel responses, which further aggravates the complexity.

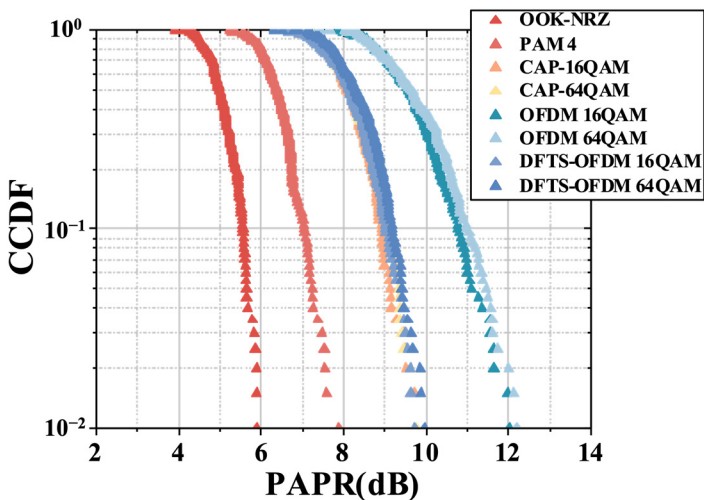

**Figure 2.** PAPR of different modulation formats in VLC.

### 2.3. Advantages and Disadvantages of ML in VLC

To show the performance benefits and costs of machine learning in visible light communication, we simply construct a visible light simulation model based on [19,25,30], as shown in Figure 3. The second-order Volterra series is used to replace the one single tap time-discretization, which represents the memory rate equation. The conversion curve of voltage, current, and optical power is used to represent the memoryless optical transform. *exp* is used as the overall channel frequency response. It should be emphasized that this is only a simplified simulation model; some parameters are determined by some communication experimental data. This simulation channel is not the focus of this article, but it is enough to illustrate the characteristics of machine learning.

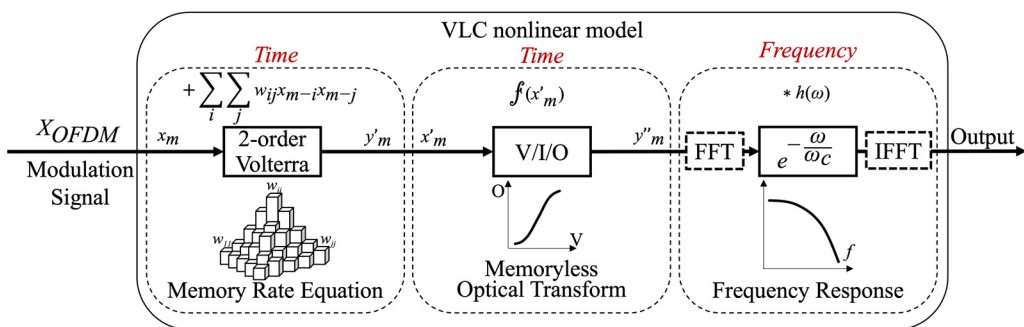

**Figure 3.** Block diagram of VLC simulation model.

In order to visualize the performance of machine learning, we chose to compare it in the field of channel equalization. Figure 4 shows the performance difference between no channel equalization, traditional nonlinear equalization, and machine learning. We used the least mean square (LMS)-based second-order Volterra algorithm as a representative of the traditional nonlinear equalizer. Both linear and nonlinear taps were set to 31. A one-layer hidden-layer multilayer perceptron (MLP) was used as a representative of machine learning. The size of the input layer was 31 and the size of the hidden layer was 128. As demonstrated in the figure, both nonlinear algorithms can have a good performance improvement. MLP outperforms the Volterra algorithm at various SNR. However, at low SNR, the enhancement is not as much, which is because this is an additive noise-limited system at this point. At high SNR, the MLP's ability to compensate for nonlinearities is even more evident.

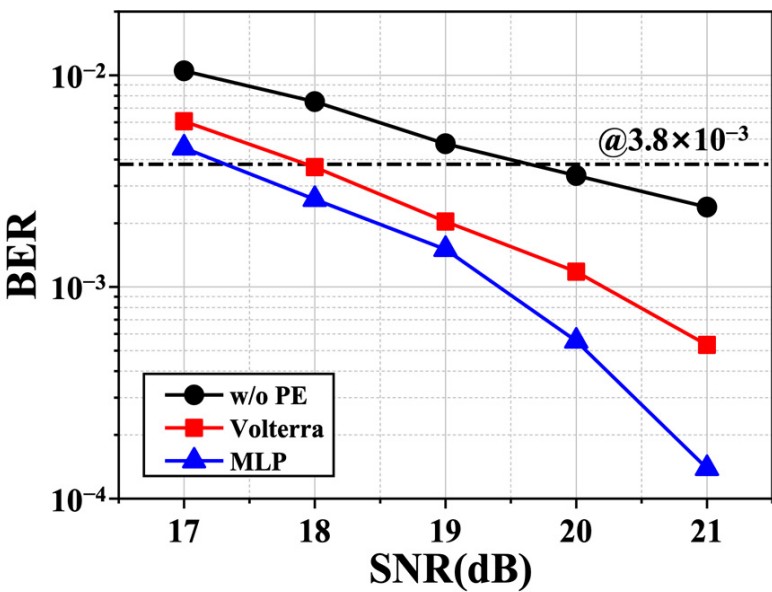

**Figure 4.** BER performance versus SNR.

As indicated by the above results, machine learning does improve performance. However, we also need to consider its drawbacks. The first is the computational complexity, as shown in Table 2. In this simulation, the trainable parameters of MLP reach 4225, while it is 527 for the Volterra algorithm. MLP has a much higher computational complexity than Volterra, which will be more obvious in actual systems where nonlinearities are more severe. This also means that machine learning algorithms need more convergence time.

**Table 2.** Computational complexity of MLP and Volterra.

| Algorithm | Input Layer | 1st Weight Layer | 2nd Weight Layer | Drd Weight Layer | Trainable Parameters |
|---|---|---|---|---|---|
| **MLP (general)** | $N$ | $W_1$ | $W_2$ | $W_D$ | $(N+1) \times W_1 + \sum_{i=2}^{D} (W_{i-1}+1) \times W_i$ |
| **Volterra (general)** | $N_{lin}$, $N_{non\text{-}lin}$ | / | / | / | $N_{lin} + \frac{N_{non\text{-}lin} \times (N_{non\text{-}lin}+1)}{2}$ |

Another problem with machine learning is generalizability. Figure 5 shows some results of generalizability studies on the Volterra algorithm and MLP. It can be seen that different training data lengths affect the performance of the equalizer. If the trained model is used for other random seed-generated data, the performance is degraded by a specific degree. This degree of degradation is relatively less in the Volterra algorithm. As mentioned above, there are many different modulation formats in visible light communication. For this reason, we also tried to use the model trained based on OFDM signals for CAP signal recovery. In this respect, one can see the more serious generalizability problems of MLP. While machine learning has better performance, data-driven learning can cause it to learn features that do not belong to the channel, for example, the data stream itself. The issues mentioned above can significantly slow down the application and development of machine learning in intelligent visible light communication. Much work should be carried out to address these aspects.

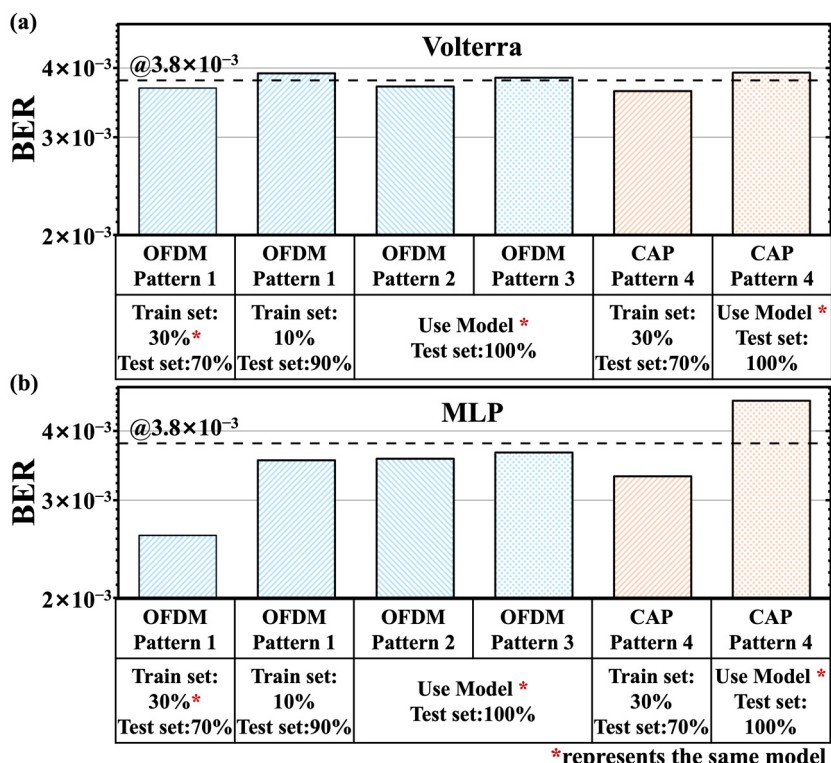

**Figure 5.** Generalizability of (**a**) Volterra and (**b**) MLP for different cases; * represents the same model.

## 3. Machine Learning in Physical Layer of IVLC

In this section, we will present some applications of machine learning in the physical layer of intelligent visible light communications, such as channel emulator, channel equalization, optimal decision, MIMO, and optimal coding, as shown in Figure 6.

### 3.1. Channel Emulator

As discussed in Section 2, the end-to-end channel for visible optical communication is exceptionally complex. In the transmission model, for example, in atmospheric environments, gas molecules and aerosol particles in the atmosphere absorb and scatter light radiation in the near-infrared band, resulting in a loss of signal received power. In addition, the change of atmospheric turbulence causes severe distortion to the optical signals. For another example, in the underwater environment, the attenuation of underwater light depends on the wavelength, where the attenuation of the signal increases with frequency. Moreover, there are other propagation effects such as temperature fluctuations, salinity, scattering, dispersion, and beam steering. For underwater VLC applications whose bandwidth is not too high (tens of MHz), the power attenuation with frequency can be approximately modeled as a linear relationship, allowing the modeling of underwater VLC multipath channels using compressive sensing (CS) method [43]. Traditional methods for high-speed point-to-point VLC cannot support accurate VLC end-to-end channel modeling, but machine learning is able to simulate the complicated nonlinear dynamics of VLC channels [44]. In massive multiple-input multiple-output (m-MIMO) VLC, the machine learning-based methods enable accurate estimation of the channel matrix [45].

### 3.1.1. TTHNet

Conducting an experimental transmission test in an underwater environment is costly, but there is no accurate analytic model as a reference for underwater high-speed VLC. In order to reduce the cost of testing underwater VLC systems, a machine learning method is needed to model the underwater channel. The two-tributaries heterogeneous neural network (TTHnet) uses a convolutional neural network (CNN) for modeling the linearity

of the underwater VLC channel and a two-layer MLP with a hollow layer for modeling the nonlinearity of the underwater VLC channel [44]. The two-branch heterogeneous structure makes full use of the CNN's shared parameters, thus reducing the system complexity. At the same time, it utilizes the MLP's extremely strong nonlinear fitting capability to fit the nonlinearity in the channel. Experiments show that the channel modeled by TTHnet is extremely close to the real channel, and the average spectrum mismatch is only 36.2% of the MLP-based channel emulator and 44.3% of the CNN-based channel emulator.

### 3.1.2. FFDNet

Since the modulation bandwidth of a single LED is limited, the use of m-MIMO LED and PD arrays are expected to substantially increase the capacity and transmission rate of VLC systems. However, due to the complexity of VLC channels, it is extremely difficult to estimate the m-MIMO channel matrix, which requires deep learning methods. Fast and flexible denoising convolutional neural network (FFDnet) is used for channel estimation in millimeter-wave communication recently [46,47], which is also applicable in VLC [45]. As an image denoising tool using machine learning, FFDnet is able to recover the input noisy channel matrix into an almost noiseless channel matrix. Compared with the minimum mean square error (MMSE) method, the FFDnet has a stronger denoising effect, which can increase the peak signal-to-noise ratio (PSNR) of the recovered channel matrix image. Unlike the nonlinear channel modeling in point-to-point high-speed VLC links, the channel matrix is treated as an image and processed using machine learning methods of image processing, which is of great importance in channel estimation of m-MIMO-VLC channels.

### 3.1.3. Conclusions

The channel capacity determines the upper bound of the communication system rate, and therefore, the accuracy of the channel estimation determines the communication efficiency of the actual system. Complex VLC channels should be accurately predicted thanks to the widespread use of powerful ML techniques in channel estimation. ML algorithms will guide IVLC to break through its own bottlenecks and complete the comprehensive integration of high-speed communication and large-scale heterogeneous networking to achieve technical solutions for next-generation communication.

### 3.2. Channel Equalization

Channel equalization techniques generally estimate the transfer function of communication channels and try to remove the channel distortion by an adaptive filter [48]. However, the common equalizers with linear adaptive algorithms become powerless in the field of high-speed VLC, because of the intrinsically limited modulation bandwidth of LEDs [49] and nonlinear distortion introduced by photoelectric devices and VLC channels. Recently, ML-based equalizers, such as artificial neural networks (ANN) [50], etc., have been developed for VLC systems. ML-based equalizers have shown outstanding equalizing performance, especially on modeling nonlinear phenomena, by adopting neural-network-based algorithms. Despite this, challenges such as massive computational complexity, slow convergence speed, and relatively poor generalization still prevent the further practical application of ML-based equalizers for VLC systems. Therefore, researchers have developed many variants, as presented next, to overcome those challenges.

### 3.2.1. Pre-Equalization GK-DNN

Conventionally, one would replace postequalization with pre-equalization to reduce the computational complexity and power consumption at the receiver side. Research works such as a weighted lookup table (WLUT), etc., have been proposed to mitigate the nonlinear distortion in VLC systems [51]. However, LUT-based pre-equalization methods suffer from a massive increase in computational complexity when dealing with high-order and high-ISI communication scenarios. Therefore, researchers have come up with ML-

based pre-equalization methods in the field of VLC systems to provide a new way of solving computational problems of LUTs.

In [52], a pre-equalization method, namely Gaussian kernel-aided deep neural network pre-distortion (GK-DNN-PD), is proposed for a high-order modulated high-speed VLC system. GK-DNN-PD outperforms the LUT-PD in terms of memory depth (MD) and the required training dataset, which leads to lower computational complexity. The experimental results show a 1.56 dB Q-factor gain compared with LUT-PD.

The proposed GK-DNN-PD method consists of two phases: the training phase and the communication testing phase. In the training phase, the received signal, which is not pre-distorted, will be linearly equalized, giving us the label sets of the GK-DNN channel estimator. Then, the clean transmitted signal with certain MD would be the feature sets. Then, the GK-DNN channel estimator will be trained to obtain the weight and bias of the estimator. Next in the communication testing phase, the weight and bias obtained in the first phase would be used to pre-distort the clean signal that is to be transmitted. Specifically, the difference between the clean signal and the output of the GK-DNN channel estimator is also considered, in addition to the weight and bias during the pre-distortion progress. Additionally, clipping operation is also adopted to reduce the peak to PAPR, which consequently reduces the nonlinear degradation.

Moreover, an NN-based pre-equalizer is proposed in [53] to mitigate the semiconductor optical amplifier (SOA) pattern effect for 50G PON, confirming the feasibility of NN-based pre-equalizer in intensity modulation and direct detection (IM/DD) system.

### 3.2.2. Postequalization GK-DNN

Since the conventional nonlinear postequalization methods based on the Volterra series suffer from a massive increase in computational complexity when dealing with high-order nonlinearity, researchers have turned to the ML for new inspirations. However, the time-consuming training progress of most ML-based postequalizers limits its actual application. To accelerate the training processing and greatly relieve the computational complexity of the equalizer at the receiver side, researchers have proposed the Gaussian kernel-aided deep neural network (GK-DNN) [54] in the field of VLC systems.

Compared to the classical MLP, the major unique feature of GK-DNN is that the input data would go through a functional mapping that is based on Gaussian function, namely the Gaussian kernel, which maps the windowed input data to a nonlinear space to reduce the number of iterations and time consumption of the fitting progress. The researchers believe that the adjacent symbols' influence towards the central (or current) one is in accordance with Gaussian distribution, hence the mapping operation would accelerate the training processing. The expression of the Gaussian kernel is given in [54]. It should be noted that the scope-controlling parameter of the Gaussian kernel would greatly affect the equalization performance of GK-DNN. Generally, the larger the parameter is, the faster the training process would be. However, there is a trade-off between the training process acceleration and equalization performance. Therefore, the Gaussian kernel parameter selection is vital to obtain the best performance. Moreover, the selection of the number of hidden layer nodes is equivalently important, which directly decides the computational complexity of the equalizer. According to the experimental results in [54], the GK-DNN equalizer could efficiently realize the postequalization in the VLC system with the aid of Gaussian kernel, which reduces the iteration epochs of the neural network by 47.06%.

### 3.2.3. Postequalization FSDNN

The frequency-slicing deep neural network (FSDNN) is a variant application of DNN that could be used in a high-speed VLC system [55]. It has the characteristics of processing high and low frequency respectively to decrease computation complexity by 11.15% compared to the traditional MLP when it comes to the equalization performance in VLC system.

In order to solve the nonlinear frequency spectrum fading issue of the received signal after going through the VLC channel, DNN is introduced as an outstanding postequalizer

to equalize linear and nonlinear distortion. However, the DNN structure must be complex enough, which means that more layers and nodes are needed and computation complexity improves to handle complicated linear and nonlinear distortions. For the expectation to release the pressure of DNN, it is worth noticing that high and low domain frequency suffer different degrees of fading. The high-frequency spectrum suffers more serious amplitude attenuation, while the low-frequency spectrum suffers less fading in the received signal in VLC system, so complex MLP structure is unnecessary for the low-frequency domain. Therefore, the received signal can be separated into high-frequency and low-frequency domains and processed, respectively, using a DNN equalizer with different complexity.

The received wide-band signal is split into two narrow-band parts in the frequency domain. Its frequency spectrum is separated into two sub-bands using a low-pass filter and a high-pass filter. Then, the two sub-band signals are respectively fed into two MLPs to train individually. The main factors of the two-MLP network should be tested artificially and adjusted to optimal values, including the number of layers, nodes in every layer, taps, and epochs. Once the MLP is finished training and the weight values are fixed, the sum of the output signal from two MLPs is the equalized and recovered signals.

### 3.2.4. Postequalization TFDNet

The commonly used ML-based equalizers in VLC systems often aim at fitting the waveform of the transmitting signal, which is a time-domain-serial signal. It is expected that the well-learned received signal should have the same spectrum as the transmitted one. However, waveform-fitting ML equalizers would sometimes cause the spectrum difference between the equalized signal and the original one. This suggests that we should take both time- and frequency-domain information into consideration to obtain a better equalization performance.

A novel postequalizer, namely joint time-frequency deep neural network (TFDNet), is reported in [56] to compensate for the nonlinear distortions in the VLC system. TFDNet could reveal comprehensive information of nonstationary signals received in the VLC system by considering both time and frequency domain information simultaneously. TFDNet can be divided into three main procedures: (1) the received one-dimensional (1D, time domain) signal goes through a short-time Fourier transformation (STFT) operation and would be transferred into a two-dimensional (2D, time-frequency domain) signal, which is a matrix and could be denoted as Y; (2) then, the obtained STFT matrix Y is fed into the NN to be trained. The labels could always be obtained by manipulating the original transmitting signal. If we assume that each row of Y represents a certain frequency component, then Y would be fed into the following network column by column; (3) finally, after the NN finishes the training progress, the reconstructed transmitting signal could be obtained by carrying out the inverse STFT (ISTFT) operation, where the analysis window must satisfy the COLA constraint [57]. Experimental results in [56] also confirm that the proposed TFDNet could resist severe nonlinear distortions and achieve a 0.1 Gbps and 0.2 Gbps data rate gain for VLC system compared to other nonlinear compensators such as Volterra and DNN.

### 3.2.5. Postequalization DBMLP

To further improve the utility of NN equalizers, researchers had proposed a modified double-branch multilayer perceptron (DBMLP) postequilibrium algorithm [44] to further reduce the consumption of energy and computational resources. DBMLP reconstructed the MLP postequalization algorithm using the structure of the Volterra series postequalization algorithm as a template. DBMLP combines the advantages of linear adaptive filters and MLP, which can improve the BER performance of the algorithm while reducing the complexity of the algorithm by 74.1%. The core structure of DBMLP is two branches of linear and nonlinear ones. In the DBMLP structure, a CNN with a convolutional layer and a dense-layer structure to simulate the linear distortion in the signal bandwidth is the first branch. In addition, a hollow MLP with an airlift layer and two dense-layer structures

to simulate the nonlinear distortion outside the signal bandwidth is the second branch. The nonlinearity of the output of the first branch is corrected by the output of the second branch, and the hollow layer can ignore the effect of the intermediate signal on the signals on both sides.

To further reduce power consumption and complexity, a pruning algorithm based on DBMLP is proposed [58]. The algorithm performs the operation of pruning by setting the smaller absolute value of weights of the connections to be pruned to 0 based on sparsity. The weights of the linear branch are not prunable while the nonlinear ones are prunable. The experimental results confirm the superiority of this approach.

### 3.2.6. Post-Equalization PCVNN

To improve the SNR in Underwater Visible Light Communication (UVLC) system, high LED power must be encouraged due to the LED's incoherent characteristic and the water medium's considerable attenuation. The nonlinearity grows more severe as the signal amplitude increases. Consequently, symbols on the outside of the constellation sustain a more nonlinear distortion than those on the inside. Based on complex-valued neural network (CVNN) [59], an adaptive partition equalizer (PCVNN) [60] has been presented, which reduces the complexity and has superior performance.

In PCVNN, the constellation is segmented into two areas by a proper threshold to distinguish between large-amplitude signals and small-amplitude signals. Then, the large- and small-amplitude signals are fed into two complex-valued neural networks. Finally, a fully connected neural network is then used to combine the signals into a complete one. Since large and small signals experience different nonlinear impairments, such a network structure can recover the signal more accurately and can greatly reduce the complexity of the model for small signals. The final experimental results also verified this conjecture [60]. PCVNN achieves up to 56.1% computational complexity reduction compared with the standard CVNN at the same performance.

### 3.2.7. Postequalization LSTM-Equalizer

High-speed VLC is limited by inherent nonlinear effects. Linear equalizers with limited taps seem powerless, and the Volterra series schemes suffer from high computational complexity when the high-order taps are required. With the rise of ML in solving nonlinear problems, long short-term memory (LSTM) networks are studied for VLC systems.

In [61], researchers proposed a memory-controlled LSTM NN equalizer for both linear and nonlinear compensation, which outperforms the conventional Volterra-based and FIR-based equalizers. LSTM carries out channel equalization as a pattern classifier where the output of LSTM cells is activated by a specially designed function. Training data with high priority would be assigned by LSTM to the latest training sequence. The proposed LSTM equalizer in [61] contains an input layer, a logical hidden layer with long and short-term memory, a classification layer, and an output layer with a merge node. A standard LSTM cell structure is used for long/short-term memory links. Moreover, a batch random resequencing procedure is adopted to control the memory effect.

Recently, the variants of LSTM have also drawn the attention of researchers because the simple LSTMs have a slow convergence speed. This is because the LSTM unit's inner parameters prolong the training period. A convolution-enhanced LSTM (CE-LSTM) equalizer, which extracts the features by using a convolutional layer, is proposed in [62] to shrink the complexity of the LSTM network and speed up the convergence progress. The experimental results also confirmed the feasibility of the proposed CE-LSTM equalizer.

### 3.2.8. Postequalization MPANN

Although the ML-based equalizers for mitigating both the linear and nonlinear distortions in VLC systems have been booming recently, the computational complexity is still a problem that needs to be further solved. Therefore, an ML-based equalizer with relatively optimal equalization performance while still maintaining a low complexity is needed in

the field of VLC. One promising way is to greatly relieve the equalizer's complexity by moderately sacrificing partial performance.

Researchers have developed a simplified ML-based equalizer, namely the memory-polynomial artificial neural network (MPANN) [63], to prune the network structure and still maintain similar equalization performance as MLP or other NNs. Likewise, the input data to be fed into MPANN could be obtained by windowing the received time-serial signal. The length of the window is usually called the memory length, which also represents the dimensions of the features. The major characteristic of MPANN is that its input layer, namely the memory-polynomial layer (MP layer), would expand the input features by one certain function, which is memory polynomial expansion. In addition, the Gaussian, Fourier basis, and other trigonometric polynomials (e.g., Legendre, Chebyshev, etc.) could be the function in the input layer. It is believed that the demanded nodes of the modified NN structure could be significantly decreased if one could provide a prior knowledge of the nonlinear model. Therefore, the memory polynomial expansion is adopted to map the input features to higher dimensional data space. Then the output pattern of the MP layer is multiplied by the corresponding weights and fed into the following hidden layer of the NN. A regular activating (ReLU) and weighting process are conducted in the hidden layer and back propagation (BP) algorithm is utilized to update the parameters. Then, finally, the output layer is utilized to output the equalized symbol. The experimental results confirmed that the MPANN could achieve the same equalization performance as the regular MLPs and only requires less than a quarter of the complexity [63].

### 3.2.9. Conclusions

As can be seen from the above presentation, the application of neural networks in channel equalization has become more than a simple application. The integration of neural networks with communication systems is starting to emerge. Figure 7 illustrates the existing neural network channel equalization in VLC. Different branches of neural networks are beginning to emerge, and many more choose to extract communication-specific features from the input data. Beyond that, fast development of computational power resources make it promising to implement ML-based modules in the field of VLC. ML-based methods with powerful nonlinear phenomenon modeling ability open a new gate to solving the inherent nonlinear problems in VLC system. However, further optimization and improvement would be needed for those ML-based equalizers in terms of computational complexity, convergence speed, and generalization. Table 3 compares the equalizers mentioned above.

**Table 3.** Summarization of machine learning algorithms for channel equalization.

| Equalizers | GK-DNN | FSDNN | TFDNet | MPANN | DBMLP | PCVNN | LSTM |
|---|---|---|---|---|---|---|---|
| Main types of NN | MLP | MLP | MLP | MLP | MLP | MLP | RNN |
| Number of hidden layers | 2 | 1 | 1 | 1 | 1 | 1 | 1 |
| Activation function | ReLU | ReLU | ReLU | ReLU | Tanh | ReLU | Tanh, Sigmoid |
| Optimizer | Adagrad | Adam | Adam | Adam | Adam | Adam | Adam |
| Complexity | Moderate | Low | High | Low | High | Low | High |
| Convergence speed | Fast | Moderate | Moderate | Moderate | Slow | Slow | Slow |
| Pre-equ. | ✓ | | | | | | |
| Post-equ. | ✓ | ✓ | ✓ | ✓ | ✓ | ✓ | ✓ |
| Deployment location | Waveform | Waveform | Waveform | Waveform | Waveform | Symbol | Symbol |

### 3.3. Optimal Decision

After channel equalization, a constellation decision is required to recover the original data. The most common decision scheme is based on the Euclidean distance between the received symbols and the standard constellation points, because the decision scheme is supposed to have the best performance in the additive Gaussian white noise (AWGN)

channel. However, as mentioned in Section 2, the VLC channel is not a simple AWGN channel, but has nonuniform noise distribution and nonlinear effect. As a result, the distortion of the constellation diagram may not exhibit a uniform Gaussian distribution around the standard constellation points. On the contrary, the constellation clusters may exhibit deviation or exhibit distortion highly relevant to the signal power. Apparently, the conventional decision is not the optimal decision scheme, and novel algorithms that take the statistical characteristics into account are required. Machine-learning-based decision schemes have been widely investigated in visible light communication systems and great improvement has been reported. These schemes can be divided into two categories: classification and clustering.

### 3.3.1. K-Means

K-means is a common unsupervised ML algorithm, which is used for spherical clustering. The main idea of K-means is to update the centroids of the constellation clusters, and the class of the new input data depends on the Euclidean distance between the input data and the dynamic centroids of the constellation clusters. It works especially well when the constellation clusters exhibit overall deviation. K-means has been applied in multiband carrierless amplitude and phase (CAP) VLC systems; experimental results indicate that for each sub-band, a decision based on K-means achieves a 1.6–2.5 dB Q-factor gain compared to a conventional scheme [64]. Moreover, if the deviation is known at the transmitting end, K-means-based predistortion is also proposed [65]. However, if the constellation clusters are not spherical, the performance of K-means will be decreased.

### 3.3.2. DBSCAN

Apart from the power-relevant nonlinear effect, a random jitter in the time domain is also a detrimental factor in VLC systems. Decision schemes based on Euclidean distance or K-means ignore the chronological order of the received sequence, so these methods are not suitable to deal with random jitter. Meanwhile, density-based spatial clustering of applications with noise (DBSCAN), as one of the clustering unsupervised ML algorithms, can divide clusters according to density, and thus has great potential in mitigating the impairment from random jitter. In [66], a DBSCAN-based decision scheme is demonstrated in VLC systems. The received one-dimensional sequence with random jitter is converted into a two-dimensional sequence with the time-axis. The key point of applying DBSCAN in the VLC system is the normalization of amplitude and the time-axis, because it is closely relevant to the density. Experimental results prove that the sequence with random jitter can still be divided into the appropriate cluster according to the density of the two-dimensional sequence.

### 3.3.3. GMM

The Gaussian mixture model (GMM) refers to decomposing a complex probability density function into several Gaussian probability density functions. In brief, GMM can make use of the linear combination of several single Gaussian probability density functions, so that the model can fit a more complex probability distribution that cannot be described by a single Gaussian function. Theoretically, if GMM contains enough Gaussian probability density functions and the weight is set reasonably, the model can fit samples with an arbitrary distribution. In a low-order modulation VLC system, the clustering algorithm deals with nonlinear problems in the constellation decision of the received signal. However, in a high-order modulation VLC system, the nonlinear effect is more obvious. When it comes to strong nonlinearity, the constellation points on the outer ring may not be a regular circular distribution. In [67], GMM is used to cluster the observation vectors formed by continuous symbols to obtain the distribution relationship between continuous symbols. The traditional soft-decision or hard-decision algorithm will directly remove the correlation between symbols, leading to linear and nonlinear damage, which will result in the lack of information leading to system performance degradation. When the correlation

between adjacent symbols is taken into account, the GMM system achieves 1 to 1.5dB sensitivity improvement. The more continuous symbols are considered, the more obviously performance improves.

### 3.3.4. SVM

The support vector machine (SVM), as one of the classical supervised ML algorithms, is usually used for classification. Through a small amount of training data, SVM can find the optimum classification plane between two clusters, and the classifier model only depends on several support vectors. If the data are not linearly separable, a kernel trick can be used for nonlinear classification. Although SVM was originally used for binary classification, multiclass SVM strategies such as one-versus-one (OVO) and one-versus-all (OVA) have been proposed. In [68], SVM-based detection is proposed and demonstrated in VLC systems. Experimental results indicate that the SVM-based scheme has a 35% increase compared with the conventional decision scheme when there is a strong nonlinear effect. In [69], a constellation decision based on SVM is investigated in integrated optical fiber and VLC systems when there is random phase rotation.

### 3.3.5. ANN

The artificial neural network (ANN) is an alternative ML algorithm for classification. The input of the ANN-based classifier is not limited to the in-phase (I) and quadrature (Q) components of the present point, whereas the I/Q components of the adjacent points in the time domain can also serve as features. The output of the ANN-based classifier is the estimated label. In this context, an ANN-based classifier serves as the nonlinear mapping process. In [70], ANN is used for the 8-color-shift keying (8-CSK) decision in an RGB-LED VLC system, and other ML algorithms are investigated for comparison.

### 3.3.6. Conclusions

Existing research has proven the feasibility of ML-based decision schemes in VLC systems. The ML-based decision schemes are summarized in Table 4. The two clustering algorithms are unsupervised, and the computational complexity is low. However, the accuracy is usually lower than supervised algorithms. In supervised algorithms, SVM has fewer computational complexity than GMM and ANN. However, the computational complexity of SVM and GMM will increase when the modulation order becomes higher. ANN is supposed to have better performance as more features can be applied for nonlinear mapping. In the future, through pruning and prior channel knowledge, the complexity of ML-based decision schemes can be further reduced.

**Table 4.** Summarization of machine learning algorithms for optimal decision.

| Algorithms | Supervision | Computational Complexity | Application |
|:---:|:---:|:---:|:---:|
| **K-means** | N | Low | Low nonlinearity |
| **DBSCAN** | N | Low | Time varying |
| **GMM** | Y | High | Moderate nonlinearity, ISI |
| **SVM** | Y | Moderate | Moderate nonlinearity |
| **ANN** | Y | High | High nonlinearity |

### *3.4. MIMO*

The multiple-input multiple-output (MIMO) technique has been developing in the field of VLC recently. Imaging MIMO and nonimaging MIMO are the two main types of MIMO-VLC systems [71]. The channel matrix is diagonal and of full rank for imaging MIMO scenarios; thus, a strict alignment is required between every LED and corresponding PD. In the nonimaging MIMO scenario, the signals will leak into each other and generate interchannel interference (ICI). Hence, to separate the mixed signals, algorithms are required. Conventional methods based on successive interference cancellation (SIC) rely on

the power proportionality of transmitters and require transmitting diversity occasionally. Moreover, the interference is most likely to be nonlinear in the MIMO-VLC system due to the optoelectrical devices. Therefore, researchers have turned to the lately booming ML for new inspiration to compensate for the ICI and nonlinearity simultaneously.

### 3.4.1. ICA

Researchers have proposed an ML-based method in [72], namely joint IQ independent component analysis (ICA), to settle spatial multiplexing problems in VLC MIMO system and enhance spectral efficiency (SE).

The model in [72] is a 2 × 2 MIMO-VLC system where superposed signals are generated. A 16-quadrature amplitude modulation (16-QAM) signal is transmitted at the Tx1 and a quadrature phase-shift keying (QPSK) is transmitted at the Tx2. As the power ratio of two Txs—namely the scaling factor—changes, the superposed constellation (SC) could be different. Blind source separation (BSS) using ICA could be adopted to separate and recover the two independent data streams from two Txs. ICA assumes that the subcomponents that compose the mixed observed signal are non-Gaussian and are statistically independent of each other. Moreover, the observed mixed signal is assumed to be the linear combination of the source signals. The proposed 2 × 2 MIMO-VLC model with different SCs just meets those assumptions mentioned above. If we assume the source signal matrix as s, the observed matrix as x, and the mixing matrix as A, where x = As, then we can obtain the recovered signal by searching for the unmixing matrix W that can linearly transform x (which is whitened) so that the estimated subcomponents are independent of each other: s = Wx. The goal of ICA is to find the unmixing matrix W which is approximately equal to the A-1. It should be noted that the mixing matrix A is of full rank. Two mixed time-domain signals in MIMO CAP-modulated system could not be separated since they share the same pulse-shaping filter pairs and consequently lost the mutual independence features at every time slot. Therefore, the joint IQ ICA method deals with the SCs at the receiver side.

### 3.4.2. MIMO-MBNN

MIMO-MBNN is featured in a hybrid structure that combines both two linear and one nonlinear equalizer to improve the received signal quality. It further removes the nonlinear loss using a NN network, which enables this equalizer to work within high nonlinearity. Although DNN could provide a powerful fitting function, its training consumes considerable computation compared with LMS and Volterra. Previous work [73] has shown that MIMO-MBNN outperforms SISO-DNN and SISO-LMS in operation range (2.33 times the area) and refreshed the record (2.1Gbps within 7% FEC) of communication rate in SR-MIMO (single receiver MIMO) VLC, demonstrating a '1 + 1 > 2' effect.

Using the SR-MIMO system in [73] as an example: the system faces both ISI and ICI; therefore, the equalizer has to remove them both. Firstly, the MIMO signal is arranged in fixed length to train the equalizer. The two linear branches deal with the linear ISI within the corresponding single channel. In the meantime, a nonlinear branch imports the training vector from both channels and uses an NN to fit the nonlinear function including the ISI and ICI. The NN outputs an R^2 vector; each element is specified for a single channel. Next, the output from a linear branch is mixed with the corresponding output from the nonlinear branch. The mixed result is the final output. As a supervised learning process, the output result is compared with the label. An Adam optimizer updates the weight in both linear and nonlinear branches. This combined structure successfully utilizes the strength of the linear equalizer and nonlinear NN network and avoids their weakness. Underwater VLC or long-haul optical communication systems could be benefited from this algorithm that improves their robustness against nonlinear loss and power jittering.

### 3.4.3. Joint Spatial and Temporal ANN Equalizer

Conventionally, MIMO decoding algorithm and compensator-like decision feedback equalizer (DFE) are adopted in MIMO-VLC receivers to compensate the spatial crosstalk

and remove the ISI step by step. The proposal of the vertical Bell Labs layered space time (V-BLAST) paves the way for joint spatial and temporal equalization in MIMO systems [74].

Considering the inherent nonlinear feature of MIMO-VLC systems, researchers have proposed the joint spatial and temporal ANN equalizer in [75] for both imaging and nonimaging MIMO-VLC links. The structure of the joint spatial and temporal ANN equalizer is similar to a matrix DFE structure. The data structure being fed to ANN contains the received signal vector with a feedforward delay line and the estimated signal vector with a feedback delay line. As for the input layer of the ANN, it contains two dimensions where one is spatial and the other is a temporal dimension. The number of input nodes is slightly adjusted to the oversampling factor of the fractionally spaced equalization scheme. According to the experimental results in [75], the proposed ANN-based joint spatial and temporal equalization scheme could outperform the traditional DFE and is able to compensate for the nonlinear channel distortion and cross-talks. Additionally, the proposed method could have better performance when the channel is ill-conditioned.

### 3.4.4. Adaptive ANN Equalizer

Spatial complexity is a major obstacle for machine learning in VLC MIMO applications. In SR-MIMO-VLC, crosstalk between channels will lead to a significant increase in the size of the data-driven neural network [73]. The traditional MIMO-LMS can obtain the channel matrix more efficiently, although it cannot take into account the nonlinear impairment. Two adaptive ANN (AANN) equalizers are proposed to combine ANN and MIMO-LMS with an adaptive parameter [76]. The spatial complexity of the AANN can be less than 10% of MIMO-MBNN.

An adaptive algorithm determines different algorithmic processes by the power ratio of two transmitted signals. When the power ratio is close to one, the SISO ANN can be used to equalize the two signals. However, when the power ratio is out of balance, it is necessary to use MIMO-ANN algorithms, such as L-DBMLP-L (combination of MIMO-LMS, DBMLP, and MIMO-LMS) or one hidden layer MBNN (OHL MBNN). This constitutes two AANN algorithms, namely ADP L/DBMLP-L and ADP MIMO ANN. The proposed AANN can achieve the same transmission performance, but with lower spatial complexity [76].

### 3.4.5. Conclusions

In summary, the ML-based ICI and ISI cancellation methods for the MIMO-VLC system are expected to be promising due to the rapid development of computational power. ML methods such as DNNs can model the nonlinear phenomenon of VLC systems where the conventional linear methods become powerless. The future research trend for ML in high-speed MIMO-VLC systems is still mainly about spatial and temporal joint equalization, as well as compensating nonlinear effects.

### 3.5. Optimal Coding

In general, to design a communication system is to split the system into several independent concatenated modules. These modules will realize the functionalities such as source/channel coding/decoding, modulation/demodulation, pre- and postequalization. The optimization of each module is carried out independently, either based on data-driven statistics features or based on mathematical models. However, the optimization of a single module cannot guarantee the overall optimization of the end-to-end communication of the entire physical layer. An intriguing approach is the end-to-end joint optimization for the physical layer [77]. The methodology is to treat physical-layer communication as an end-to-end signal-reconstruction problem, and to apply the concept of autoencoder to represent the physical-layer communication modules (the transmitter, the channel, and the receiver) by one deep neural network. Autoencoder is an unsupervised deep learning algorithm. The goal of the autoencoder is to find an optimized representation at its intermediate layer. This intermediate representation is robust to channel perturbations, allowing the output to be reconstructed with minimal error.

The end-to-end learning method extended its applications in MIMO and OFDM cases [78–80] with satisfying results. The autoencoder establishes a unified physical-layer framework that can be used in complex communications scenarios, and can obtain a lower bit error rate than the classic method through learning with lower computational complexity. Most of these autoencoder approaches are optimized on the symbol-wise categorical cross entropy [80]. However, the communication system is defined by the bit error rate. Novel approaches developed in [80,81] can optimize the bitwise mutual information between the input and output. Apart from that, VLC systems should consider more instability of complicated nonlinear distortions when searching for the optimal coding scheme. Considering recent works of VLC-based autoencoders, substantial efforts are still demanded to develop practical solutions in the wireless optical system.

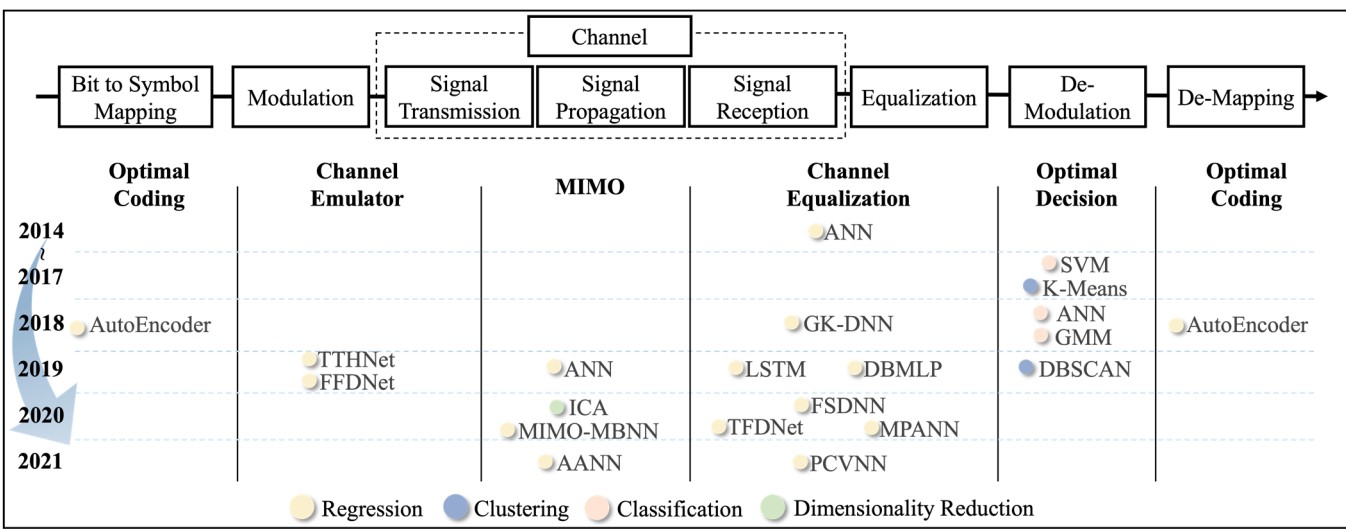

**Figure 6.** Overview of machine learning algorithms in the intelligent physical layer; Optimal coding: AutoEncoder [82–86]; Channel emulator: TTHnet [44], FFDNet [45]; MIMO: ANN [75], ICA [72], MIMO-MBNN [73], AANN [76]; Channel Equalization: ANN [50], GK-DNN [52,54], LSTM [61], DBMLP [44], FSDNN [55], TFDNet [56], MPANN [63], PCVNN [60]; Optimal decision: SVM [68], K-means [64], ANN [70], GMM [67], DBSCAN [66].

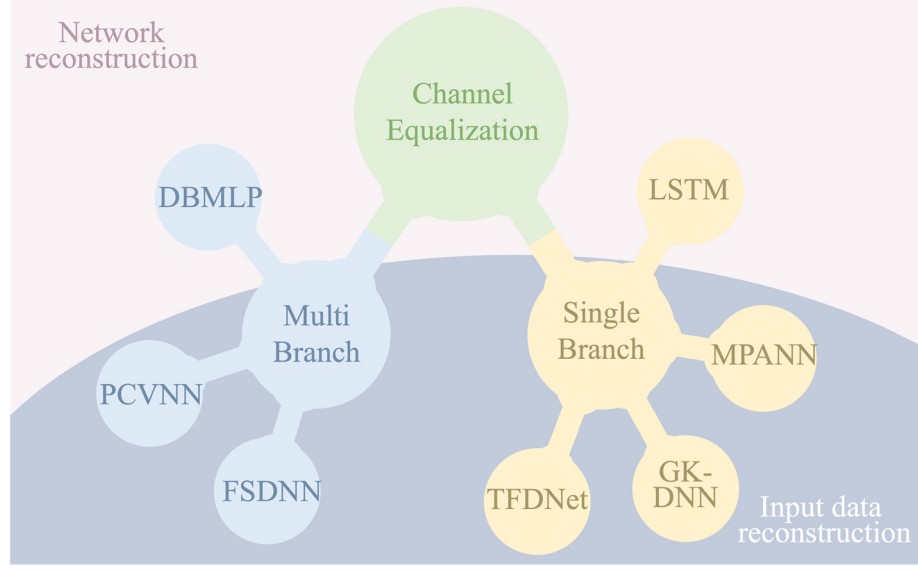

**Figure 7.** Channel equalization in VLC systems.

### 3.5.1. VLC-Based Autoencoder

End-to-end learning of the transmitter and receiver for communications over a visible light channel was first proposed in [82]. The transmitter and receiver are mapped to the encoder part and decoder part of the autoencoder structure. Symbol-level precoding at the encoder input transfers the original symbols to one-hot vectors. To accurately reconstruct the input from the received signal, the decoder works as a classifier. The transmission channel is built with two fixed weighted layers considering the color crosstalk of optical antennas and additive noise. Then, the categorical cross-entropy function is used to evaluate the loss between the input and the output probability vector of the transmitted symbols. Gradients of symbol errors direct the updating of the encoder and decoder layers through the backward-propagation process. The average symbol error rate performance reflects the superiority of the proposed closed-loop optimized transceiver to a minimum distance maximizing modulation scheme [83]. Apart from the photodiodes-based autoencoder [82,84], in [85] a convolutional autoencoder structure is proposed for image sensor communication systems. The system utilizes spatially separated LED arrays to convey an OOK-modulated signal and an optical image sensor as the receiver. Convolutional layers are implemented to overcome irradiance spread and lens blur induced by the sensor. The 2D convolution operations of the proposed autoencoder bring performance gain for image-decoding strategies in the simulated ISC systems. Besides the above works on decreasing error rates, the method in [86] takes flicker and illumination levels into account and tries to address real-life application constraints. However, all these works stay with numerical validations. As discussed in Section 2.1, it should be noticed that the channel estimation work is still in a naive stage. Experimental validation of the aforementioned end-to-end communication schemes will suffer performance degeneration due to the unconsidered dynamic nonlinear impairments of devices. To the best of our knowledge, there is still no experimental demonstration of an autoencoder-based VLC system.

### 3.5.2. Fiber/Wireless-Based Autoencoder

The parameters of the fiber channel or wireless channel are very important for end-to-end learning, so that the back-propagation algorithm of the neural network can be effectively calculated. In an optical fiber communication system, the channel is governed by the nonlinear Schrödinger equation (NLSE). Therefore, in order to apply autoencoder in optical fiber communication, an appropriate equivalent deep neural network of NLSE has to be established [87–90]. The experimental results demonstrated that the input information was mapped to a set of robust transmitted waveforms via autoencoder and detected with a measured BER under FEC threshold in intensity-modulation direct-detection (IM/DD) optical-fiber systems [87,88]. The successful demonstrations in both wireless communication and optical fiber systems clearly validate that end-to-end learning can be a promising technology to fundamentally reconsider communication-system design [77,87].

### 3.5.3. Conclusions

Above all, the only difference between VLC, fiber, and wireless-based autoencoders lies in the channel. While the output signals of fiber or wireless can be analytically modeled with prior domain knowledge and experience, predicting the outputs of a VLC channel by the mathematically convenient models is very hard, if not impossible. Hence, applying data-driven ML in VLC systems appears to be a reasonable direction. However, even the most recent works have not been practically implemented. Future works should pay attention to dealing with more realistic issues such as the low-frequency interference problem or dynamic nonlinear distortions in the VLC systems. Moreover, the current method relies on cost-prohibitive computational and temporal resources. Meta-learning-enabled online training will speed up the application of end-to-end communication links in the 6G architecture.

## 4. Future Trend of ML in IVLC

Over the last decade, visible light communication has achieved rapid advances both in technologies and in applications. The fundamental impetus to this achievement is not only the progress in communication technologies including coding, modulation, and signal processing, but also the rapid development of optoelectronic chips and devices. The bandwidth of the state-of-art VLC devices can exceed 1 GHz [91], nearly 100 times larger than that of 10 years ago. However, compared with the devices used in optical fiber communication, which usually boast nearly 100 GHz bandwidth, the bandwidth for VLC is too small to afford roughly equivalent data rates with optical fiber communication. Thus, the integration of wired/wireless communication in future ultrahigh-data-rate 6G networks could still be a challenge. Coherent light sources, i.e., lasers, are supposed to have larger bandwidths. Semipolar and nonpolar LEDs are reported to increase the modulation bandwidth. Microstructure, microcavity, and plasmon may be promising new approaches to enable ultrahigh data rates [92]. The intrinsic diversified characteristics of VLC devices require AI technologies to understand the device model, optimize the transmission link, and manipulate the whole network effectively.

The future development of IVLC in the intelligent physical layer and intelligent network layer will be presented in this part, as shown in Figure 8. We strive to provide readers with some insight.

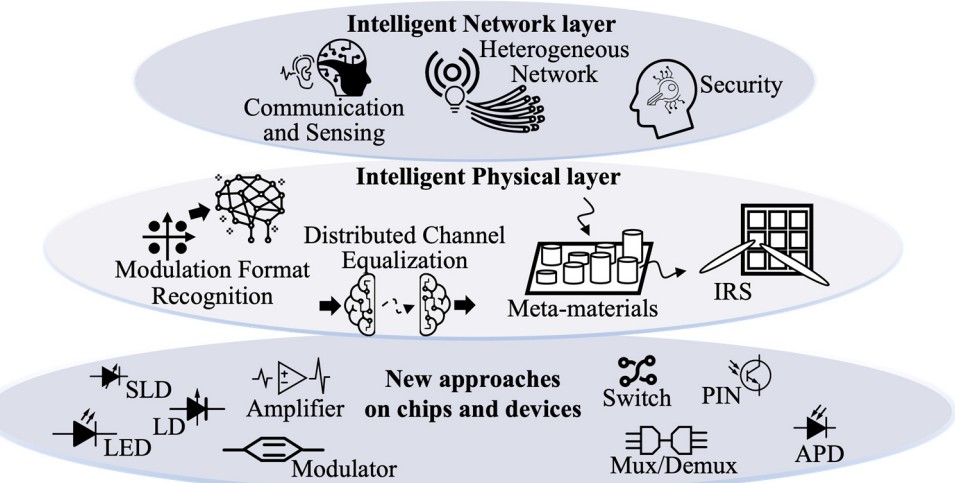

**Figure 8.** High-level view of intelligent physical layer and intelligent network layer of IVLC.

### 4.1. Intelligent Physical Layer

In this section, we will go through how electromagnetism and information theory can be better applied to AI-driven visible light communication. We will then present the possible forms of distributed learning of machine learning at the intelligent physical layer. Finally, a future application of machine learning in IVLC will be presented.

#### 4.1.1. Fundamental Electromagnetism Theory and Frontiers in Optical Physics

The frequency bands available for next-generation wireless communication evolve to higher frequency bands such as millimeter wave, terahertz, and visible light [93]. In this case, the network spectrum and resources are unprecedentedly plentiful. Efforts are underway to sense the space electromagnetic information, to govern the spectrum allocation and beamforming by combining the fundamental electromagnetism theory with the conventional information theory. Intelligent Reflecting Surfaces (IRS) is a promising research direction in mm-waves and terahertz waves [94]. In visible light communication, we can imagine the optical phased array antenna. The feature size of the photonic circuits is far larger than that of the microelectronic circuits. Metamaterials, metasurfaces, and metalenses enable manipulation of the propagation, polarization, amplitude, and phase of

the injected light at a deep subwavelength scale. This kind of optical artificially structured material is ultracompact in a scale of tens or hundreds of nanometers. On one hand, AI technologies can be a useful tool to design such kinds of materials by using optical reverse design. On the other hand, metamaterial can be a powerful way to realize optical neural networks and optical computing. All of these ideas are related to optical physics and could be promising research avenues towards 6G.

### 4.1.2. Distributed Channel Equalization

As mentioned above, existing research on machine learning in VLC has mainly focused on the standalone receiver or transmitter side. However, the available computational resources of the terminals and the central office are apparently different, so that leaving the computational complexity at the receiving or transmitting end only is not a reasonable solution. Therefore, ML-based distributed channel equalization at both the transmitting and receiving end is worthy of investigation. In [65], predistortion based on K-means is proposed to mitigate nonlinear impairment in VLC systems. A spatiotemporal neural network-based predistortion equalizer has also been proposed in RF communication systems in order to compensate for the nonlinearity caused by the power amplifier [95]. CNN has also been proposed to be applied for behavioral modeling and digital predistortion, and the model's coefficients can be significantly reduced [96]. Both NN-based distortion and postequalization do promise great performance.

However, an essential premise for pre-equalization is that the channel information state (CSI) is known to the transmitter. The CSI is usually obtained by training sequence. Unfortunately, the VLC channel is usually not static, and there is a mismatch between the estimated CSI and the actual CSI, as the statistic characteristics of the transmitted signal may be changed after pre-equalization or predistortion. The entire channel equalization should be computed simultaneously at the transmitter and receiver side in a distributed manner, taking into account the computational resources of both the receiver and transmitter as well as performance optimization. The CSI mismatch can be compensated since the training sequence is available at the receiving end. Therefore, distributed channel-equalization methods are required in the future VLC system.

### 4.1.3. Modulation Format Recognition

With the continuous improvement of the transmission capacity, the future VLC networks must be a mixture of multiple transmission rates and multiple modulation formats, thus modulation format recognition (MFR) will become an essential part of the overall communication system. In the early years, traditional machine learning methods, such as decision trees and SVM [97], were applied in VLC format recognition. The drawback of these solutions is that they rely on manual feature extraction, which leads to a lack of flexibility and portability.

Recently, deep learning is widely used in pattern recognition because of its ability to mine useful feature information at a deeper level. DNN is able to extract deeper layers of the signal by stacking layers of hidden layers [98]. However, as the number of hidden layers in DNN increases, the structural complexity increases. Therefore, it is especially important to design the network structure of DNNs rationally. CNN performs local feature extraction of information by setting appropriately sized convolutional kernels, and subsequently achieves classification through fully connected layers [99,100]. The feature-extraction part is mainly composed of convolutional and pooling layers, and the recognition and classification part is the same as the fully connected layer of the BP neural network. Other schemes that combine format recognition with deep learning, such as RNN [101], PNN [102], etc., have been gradually proposed.

It can be foreseen that due to the complexity of visible communication channels and the diversity of modulation formats, there will be an urgent need for machine learning-based modulation format technology. AI-driven MFR will effectively improve the recognition rate and accuracy, accelerating the pace of visible light communication applications.

*4.2. Intelligent Network Layer*

In this section, communication-aware integrated networks, as well as heterogeneous networks, will be introduced first. Finally, since visible light communication has the characteristics of both wireless and optical communication, network security will be introduced.

4.2.1. Converged Communication and Sensing

The convergence of sensing and communication network has become one of the leading trends in 6G technology and services [93]. The 6G network is expected to be a fusion of mobile communication, sensing, and intelligent computing. Such a converged network refers to a system that has the capabilities of target positioning (ranging, speed measurement, angle measurement), target imaging, target recognition, and target tracking. The development of higher-frequency bands such as millimeter waves, terahertz, and visible light will have more and more overlaps with traditional sensing frequency bands. Wireless communication and wireless sensing show more and more similarities in system design, signal processing, and data processing. Therefore, the AI-propelled VLC technologies will be found to be efficient in sensing technologies. A pragmatic solution is to allow communication and sensing in the same frequency spectrum. Research endeavors must be devoted to the solution of avoiding interference, and improving spectrum utilization. The codesign of the sensing and communication waveform may be aided by AI tools. The functionalities of communication and sensing are obtained based on software and hardware resource sharing or information sharing, which can effectively improve spectrum efficiency, hardware efficiency, and information-processing efficiency.

4.2.2. Heterogeneous Network

6G networks are becoming more and more heterogeneous, composed of different access standards and various network deployment methods. Therefore, it is necessary to consider the indoor and outdoor heterogeneous networking and interconnection issues of visible light communication networks and other communication technologies, for instance, the power line communications, optical fiber access networks, and mobile communications in radio frequency and even higher-frequency bands. Within this complex architecture of the future network, it is difficult for traditional models and algorithms to provide efficient and reliable technical support. AI-driven network technologies have made a series of progress in wireless communication to handle interference coordination and resource scheduling (including power allocation, channel allocation, and access control), which will reduce future wireless resource management costs and improve service quality.

The traditional network structure is network-centric with quite limited flexibility. The users are passive nodes, and the cell is generally preset to a fixed shape according to the transmission scheme and does not change with the communication traffic. Distributed artificial intelligence can implement a fully user-centric network architecture. By taking advantage of the diversity of user locations and service requirements, virtual amorphous communities can be constructed to provide better services for each user. AI can be used to understand users and perform user prediction, inference, and big data analysis. Moreover, AI can realize self-organizing network operations and management by network edge computing, and eventually form global closed-loop optimization.

4.2.3. Security Network

The characteristics of optical networks and wireless communications are combined in VLC systems. As a result, VLC suffers more complex cyber security challenges. When VLC becomes the infrastructure of future communication services, it will be subjected to additional attacks [103].

The first is the jamming attack in visible light communication. Since VLC is an LOS channel, it is highly susceptible to interference by external interference sources. Brief jamming attacks can cause enormous volumes of data to be incorrect or leaked due to the high transmission rates. ML can identify the presence and level of interference by learning

interference signals targeting the physical layer in optical communications, as an example of deep Q-learning (DQN) [103].

The second is the cyber-physical attack in visible light communication. VLC is vulnerable to external intervention because it runs in an open environment. Injecting damaging signals into unprotected visible communication equipment can have a variety of negative consequences for communication [104]. By learning the characteristics of different devices or watermarks carried by the transmission, ML can be used to identify illicit transmission devices in VLC systems, as an example of TF-FSN [105].

The third is eavesdropping in visible light communication. VLC channels are vulnerable to unauthorized terminals because of their broadcast nature. Eavesdropping in public places compromises the privacy of legitimate users. Eavesdroppers' capacity to infer information over the channel is reduced by a smart beam anti-eavesdropping system based on deep reinforcement learning (DRL) [106]. The suggested intelligent beamforming method based on DRL may also make full use of complicated and high-dimensional structure information, improving network security for users.

VLC has some fixed security properties; however, entities communicating under the same space are vulnerable to discovery and information theft. Future research will focus on how to successfully use machine learning to counter attacks in the visible light field.

## 5. Conclusions

In this paper, we have provided a comprehensive study on AI-driven intelligent visible light communication. The major challenges in visible light communication are addressed, as well as the specific contribution of AI-enabled IVLC in overcoming these challenges.

Nonlinearity introduced by VLC due to additional electro-optical conversion is one of the major challenges that can significantly impair communication performance. Because of the spontaneous radiation properties of LEDs, visible light signals can only be directly modulated in communication. This means that changes in signal amplitude will directly affect the carrier concentration and thus the recombination of electrons and holes. This also increased the difficulty of modeling the visible light communication E2E channel. The improvement of machine learning for visible light communication performance is demonstrated in the paper, but again, its drawbacks of high computational complexity and low generalizability are presented.

Detailed applications in the intelligent physical layer of IVLC are categorized into five scenarios based on the communication system framework: optimal coding, channel emulator, MIMO, channel equalization, and optimal decision. For each of these categories, detailed elaboration is given to the state of the art. Among these technologies, autoencoder has the potential to revolutionize the existing physical layer communication architecture as a means of optimizing end-to-end communication.

Finally, we envisage the prospect of the intelligent physical layer and intelligent network layer. As AI continues to integrate in optical physics and electromagnetism area, the derived IRS, metamaterials, metasurfaces, metalenses, and optical computing will further drive the development of IVLC. In particular, optical computing and optical neural networks will be the focus of development. As IVLC may be deployed in 6G on a large scale, electricity-based digital signal processing will consume a lot of energy. The use of optical computing will greatly reduce consumption. Distributed channel equalization combines the existing communication system and the multilayer mechanism of neural networks, which will be an effective means of rapid deployment of IVLC. At the network layer, the main role of AI will be to reduce human intervention. It can achieve better resource scheduling and security through intelligent learning.

Hopefully, it will be a thorough investigation of intelligent visible light communication and serve as a practical guide for large-scale deployment of visible light communications in future 6G networks.

**Author Contributions:** Conceptualization, J.S. and N.C.; methodology, J.S. and N.C.; software, J.S.; validation, W.N.; formal analysis, J.S. and W.N.; data curation, J.S.; writing—original draft preparation, J.S., W.N., Y.H. and Z.X.; writing—review and editing, Z.L. and N.C.; visualization, J.S., W.N. and Z.X.; supervision, N.C.; project administration, J.S. and N.C.; funding acquisition, S.Y. and N.C. All authors have read and agreed to the published version of the manuscript.

**Funding:** This research was funded by the Natural Science Foundation of China Project (No. 61925104, No. 62031011), the Major Key Project of PCL (PCL2021A14), China Postdoctoral Science Foundation (2021M700025), and China National Postdoctoral Program for Innovative Talents (BX2021082).

**Institutional Review Board Statement:** Not applicable.

**Informed Consent Statement:** Not applicable.

**Data Availability Statement:** The data that support the findings of this study are available from the corresponding author upon reasonable request.

**Conflicts of Interest:** The authors declare no conflict of interest.

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
