# Peer review of "AI-Enabled Intelligent Visible Light Communications: Challenges, Progress, and Future"

_photonics, doi:10.3390/photonics9080529_

Round 1

Reviewer 1 Report

The authors give a good review on the topic of AI based VLC.  (1) The equations (6)-(8) are not displayed well.  (2) Conclustions can be enhanced to give more detail results and highlights. (3) Future trend of ML in VLC can be added more viewpoints. 

Author Response

The revised manuscript (pdf) please see the attachment.

To Reviewer 1:

Q1. The equations (6)-(8) are not displayed well.

A1. Thank you for your kindly reminds. We will double check the quality of the auto-converted PDF.

Q2. Conclusions can be enhanced to give more detail results and highlights.

A2. Thank you for your useful comments. We have enhanced the conclusions in the revised version.

The paragraph Nonlinearity introduced by VLC due to the additional electro-optical conversion is one of the major challenges, which can significantly impair communication performance. Because of the spontaneous radiation properties of LEDs, visible light signals can only be directly modulated in communication. This means that changes in signal amplitude will directly affect the carrier concentration and thus the recombination of electrons and holes.  This also increased the difficulty of modeling the visible light communication E2E channel. The improvement of machine learning for visible light communication performance is demonstrated in the paper, but again, its drawbacks of high computational complexity and low generalizability are presented. is added.

The paragraph As AI continues to integrate in optical physics and electromagnetism area, the derived IRS, meta-materials, meta-surfaces, meta-lens and optical computing will further drive the development of IVLC. In particular, optical computing and optical neural networks will be the focus of development. As IVLC may be deployed in 6G on a large scale, electricity-based digital signal processing will consume a lot of energy. The use of optical computing will greatly reduce consumption. Distributed channel equalization combines the existing communication system and the multilayer mechanism of neural network which will be an effective means of rapid deployment of IVLC. At the network layer, the main role of AI will be to reduce human intervention. It can achieve better resource scheduling and security through intelligent learning. is added.

Others can be seen in the uploaded manuscript.

Q3. Future trend of ML in VLC can be added more viewpoints.

A3. Thank you for your kindly suggestion. We have added more viewpoints in the section of 5. Conclusion.

The paragraph As AI continues to integrate in optical physics and electromagnetism area, the derived IRS, meta-materials, meta-surfaces, meta-lens and optical computing will further drive the development of IVLC. In particular, optical computing and optical neural networks will be the focus of development. As IVLC may be deployed in 6G on a large scale, electricity-based digital signal processing will consume a lot of energy. The use of optical computing will greatly reduce consumption. Distributed channel equalization combines the existing communication system and the multilayer mechanism of neural network which will be an effective means of rapid deployment of IVLC. At the network layer, the main role of AI will be to reduce human intervention. It can achieve better resource scheduling and security through intelligent learning. is added.

Reviewer 2 Report

This paper provided a comprehensive study on AI-driven intelligent visible light communication. The major challenges in visible light communication are addressed, as well as the specific contribution of AI-enabled IVLC in overcoming these challenges. The prospect of the intelligent physical layer and intelligent network layer are described. In general, this is a high-quality research paper with rich introductory material and solid technical contribution. This reviewer has the following issues for the authors to consider while revising the paper:

- In Section 2.1 Visible Light Communication E2E Channel, the current challenges of end-to-end VLC channels can be summarized in a list for easy reading. For instance, you can label the challenges, reasons, related references. Specifically, on line 179, for the challenges of VLC channel modeling due to large signals and wide bandwidth, the references include [31-33].

- In Section 3. Machine Learning in Physical Layer of IVLC, it is recommended that some experimental data can be supplemented for the introduction of the application of machine learning in the area of IVLC physical layer. For example, on line 342, "Experiments show that the channel modeled by 342 TTHnet is extremely close to the real channel and is completely more accurate than the 343 deep neural network using only CNN or MLP", where "extremely close" and "more accurate" are abstract descriptions rather than concrete data. Besides, most of this section is introducing the machine learning algorithms applied in IVLC physical layer in literature, while it is lack of conclusive statements or opinions from the authors for these algorithms, such as whether some of the algorithms need improvement?

- In Section 2.1 Visible Light Communication E2E Channel, some subscripts "c" in formulas are shown as "?", such as on line 119, equations (6) and (7), and line 156. Please check. Besides, on line 172, "... to make the left side of Eq.Error! Reference source not found", some symbols in equation cannot show correctly.

-The ":" following "2020" in reference 2 should be changed to "."? Some short names of the journals are incomplete, such as "IEEE Wireless Commun," should be "IEEE Wireless Commun.,";  "IEEE J Sel Areas Commun," should be "IEEE J. Sel. Areas Commun.,"

Author Response

The revised manuscript (pdf) please see the attachment.

To Reviewer 2:

Q1. In Section 2.1 Visible Light Communication E2E Channel, the current challenges of end-to-end VLC channels can be summarized in a list for easy reading. For instance, you can label the challenges, reasons, related references. Specifically, on line 179, for the challenges of VLC channel modeling due to large signals and wide bandwidth, the references include [31-33].

A1. Thank you for your useful suggestion. We summarized the current challenges of end-to-end VLC channels in Table1. We also revised the sentence on line 179.

The Table1 is added.

Table 1. Challenges of Visible Light Communication E2E Channel.

Challenges

Reasons

References

Optoelectronic and electro-optical conversion

Introduce additional nonlinearity

[26-29]

Large signals

Bring the device into the nonlinear region

[21]

Wide Bandwidth

Introduce severe ISI

[19]

Different transmission channel modeling

Diverse application scenarios, such as indoor, underwater

[31-33]

The sentence In addition to the large signals and high bandwidth that make visible light channel modeling difficult[19, 21], another challenge is transmission channel modeling[31-33]. is revised.

Q2. In Section 3. Machine Learning in Physical Layer of IVLC, it is recommended that some experimental data can be supplemented for the introduction of the application of machine learning in the area of IVLC physical layer. For example, on line 342, "Experiments show that the channel modeled by 342 TTHnet is extremely close to the real channel and is completely more accurate than the 343 deep neural network using only CNN or MLP", where "extremely close" and "more accurate" are abstract descriptions rather than concrete data. Besides, most of this section is introducing the machine learning algorithms applied in IVLC physical layer in literature, while it is lack of conclusive statements or opinions from the authors for these algorithms, such as whether some of the algorithms need improvement?

A2. Thank you for your kindly suggestion. We have added more experimental data to each machine learning method. We also add a small conclusion in each section. The titles of these sections are “3.1.3. Conclusion, 3.2.9. Conclusion, 3.3.6. Conclusion, 3.4.5. Conclusion, 3.5.3. Conclusion”.

The sentence Experiments show that the channel modeled by TTHnet is extremely close to the real channel, and the average spectrum mismatch is only 36.2% of MLP based channel emulator and 44.3% of CNN based channel emulator. is revised.

The sentence The experimental results show a 1.56dB Q-factor gain compared with LUT-PD. is revised.

The sentence According to the experimental results in [54], GK-DNN equalizer could efficiently realize the post-equalization in VLC system with the aid of Gaussian kernel, which reduces the iteration epochs of neural network by 47.06%. is revised.

The sentence It has the characteristics that high and low frequency are processed respectively to de-crease 11.15% computation complexity compared to the traditional MLP when it comes to the equalization performance in VLC system. is revised.

The sentence Experimental results in [56] also confirm that the proposed TFDNet could resist severe nonlinear distortions and achieve 0.1Gbps and 0.2Gbps data rate gain for VLC system compared to other nonlinear compensators such as Volterra and DNN. is revised.

The sentence PCVNN achieves up to 56.1% computational complexity reduction compared with the standard CVNN at the same performance. is revised.

The sentence K-means has been applied in multiband carrier less amplitude and phase (CAP) VLC system, experimental results indicate that for each sub-band, decision based on K-means achieves 1.6-2.5dB Q-factor gain than conventional scheme [64] is revised.

The sentence When the correlation between adjacent symbols is taken into account, the GMM system achieves 1 to 1.5dB sensitivity improvement. is revised.

The sentence Experimental results indicate the SVM-based scheme has 35% increasement compared with the conventional decision scheme when there is a strong nonlinear effect. is revised.

Q3. In Section 2.1 Visible Light Communication E2E Channel, some subscripts "c" in formulas are shown as "?", such as on line 119, equations (6) and (7), and line 156. Please check. Besides, on line 172, "... to make the left side of Eq.Error! Reference source not found", some symbols in equation cannot show correctly.

A3. Thank you for your kindly reminder. We only uploaded the word file. Some errors may occur during the converting process. We will double check the quality of the auto-converted PDF.

Q4. The ":" following "2020" in reference 2 should be changed to "."? Some short names of the journals are incomplete, such as "IEEE Wireless Commun," should be "IEEE Wireless Commun.,";  "IEEE J Sel Areas Commun," should be "IEEE J. Sel. Areas Commun.,"

A4. Thank you for your kindly reminder. We have reformatted all the references. Some examples are shown below:

  1. Latva-Aho M, Leppänen K, Clazzer F, et al. Key drivers and research challenges for 6G ubiquitous wireless intelligence. 2020.
  2. Zong B, Fan C, Wang X, et al. 6G technologies: Key drivers, core requirements, system architectures, and enabling technologies. IEEE Veh. Technol. Mag., 2019, 14(3): 18-27.
  3. Chi N, Haas H, Kavehrad M, et al. Visible light communications: demand factors, benefits and opportunities [Guest Editorial]. IEEE Wireless Commun., 2015, 22(2): 5-7.
  4. Ying K, Qian H, Baxley R J, et al. Joint optimization of precoder and equalizer in MIMO VLC systems. IEEE J. Sel. Areas Commun., 2015, 33(9): 1949-1958.
